# C-S-H Seeds Accelerate Early Age Hydration of Carbonate-Activated Slag and the Underlying Mechanism

**DOI:** 10.3390/ma16041394

**Published:** 2023-02-07

**Authors:** Bo Yuan, Hengkun Wang, Dianshi Jin, Wei Chen

**Affiliations:** 1School of Material Science and Engineering, Wuhan University of Technology, Wuhan 430070, China; 2State Key Laboratory of Silicate Materials for Architectures, Wuhan University of Technology, Wuhan 430070, China

**Keywords:** carbonate activation, early age reaction, C-S-H seeds, accelerated hydration, seeding effect

## Abstract

The slow hardening process of carbonate-activated slag limits its application as a construction material. This paper aims to provide an acceleration method for the early age hydration of carbonate-activated slag by applying calcium silicate hydrate (C-S-H) seeds and unveil the underlying mechanism. The results show that the incorporation of C-S-H seeds significantly accelerates the early age reaction of carbonate-activated slag and shortens the setting time. With 4% of calcium silicate hydrate (C-S-H) seeds, the 1d-compressive strength of carbonate activates slag can achieve 25.4 MPa. The C-S-H seeds acts as the preferred nucleation sites for the strength-giving phase C-A-S-H gel and the carbonate-containing phases (e.g., calcite, gaylussite, hydrotalcite, etc.), and accelerates hydration. The dormant period of samples with C-S-H seeds becomes negligible, confirming that the seeding effect that controls the saturation limits of the pore solution is the major reason for the accelerated hydration.

## 1. Introduction

Low alkalinity, ease of operation, and good performance are the advantages of carbonate activation, which has drawn significant attention from the academic and industrial fields in recent years [1,2,3]. The long setting time caused by slow hydration is the major challenge that is still to meet the basic requirements for in situ applications [4,5]. Multiple methods have been conducted to solve this problem, e.g., mixing with high-alkaline solutions (NaOH and waterglass) [6], calcareous materials (CaO and Ca(OH)_2_) [7], and other additives such as calcined layered double hydroxides (CLDHs) [5]. A few studies also works on the early age hydration mechanism to find an explanation for the slow hardening process and to provide solutions without compromising its performance [8].

A moderate pH value and high concentration of CO_3_^2−^ in pore solution are considered the primary reasons for the slow hardening process of sodium carbonate-activated slag (SCAS) [8]. Slow pH development defers the ionic release from slag particles, and the presence of CO_3_^2−^ also restrains the level of Ca^2+^ in pore solutions, postponing the formation of calcium silicate hydrates. Incorporated extra agents increase the alkalinity of activators, promote the dissolution rate of elements of Si, Al, and Ca from slag particles, and accelerate the formation of hydration products, such as Al-substituted calcium silicate hydrates (C-A-S-H gel) and carbonate-containing phases. The hardening process of carbonate activation is accelerated; however, reductions in strength or issues of durability are also often reported [1,6,9,10]. Importantly, increasing alkalinity also raises the risk of storage, operation, and costs, limiting their wide application in in situ constructions.

C-A-S-H gel and carbonate-containing phases, e.g., calcium carbonate, gaylussite, hydrotalcite, are the main hydration products of SCAS [11]. It is worth noting that the solubility of these reaction products is similar, especially for the carbonate-containing phases. However, the precipitation of the strength-giving phase C-A-S-H gel often occurs after the formation of carbonate-containing phases, probably due to the highly concentrated CO_3_^2−^ anions introduced by the activator [12]. Consequently, the hardening process of SCAS delays. It is worth noting that the precipitation tendency of C-A-S-H gel and carbonate-containing phases not only depends on their solubility but also can be altered by additives, e.g., seeding [13,14].

Seeding is one of the most straightforward methods used to control supersaturation, and a small mass of seeds added to a supersaturation can start the crystallization process at the desired supersaturation level [15]. Calcium silicate hydrate (C-S-H) is the main hydration product and binder phase in the cementitious materials and alkali-activated slag binder; moreover, it shares a similar structure to the C-A-S-H gel [16]. The structure of C-S-H can be described using a highly deformed tobermorite structure, i.e., tobermorite with a layered structure which lacks bridged silicon–oxygen tetrahedra, resulting in long chains of silicon–oxygen tetrahedra.

C-S-H seeds are the nanoparticles of C-S-H with various Ca/Si ratios that can be synthesized using the methods of pozzolanic, sol-gel, and precipitation [17]. C-S-H seeds have been found to be efficient in stimulating the dissolution of cement clinker and portlandite, which facilities the formation of C-S-H gel in the early age hydration of cement-based materials [18,19]. The incorporation of C-S-H seeds can promote the strength development of cementitious materials without weakening their long-term mechanical performance, especially those with high replacement levels of supplementary cementitious materials (SCMs) that have lower activity compared to cement [14,15]. C-S-H seeds have been widely studied in the field of cement-based materials and have the following advantages [14,15]: (1) reduce cement consumption, help reduce carbon dioxide emissions, and protect the environment; (2) promote the early strength of cementitious materials without causing a notable negative effect on the long-term strength development; (3) exhibits better durability, longer service life, and reduces repair and maintenance costs; (4) convenient to synthesize, and C-S-H seeds can be prepared through hydrothermal synthesis from cheap raw materials such as silica and calcium oxide.

A large surface area and low interfacial energy in the hydration products of C-S-H seeds stimulate the precipitation of cement hydration at an early age and compensates for the negative effect of SCMs and additives (e.g., superplasticizers) on the reaction, leading to faster and better strength development [17,20,21,22,23]. Adding C-S-H seeds promoted the formation of the C-A-S-H gel phase and compensated the retardation effect of the polycarboxylate superplasticizers on the cement slurry [19,21]. The separation of the hydration products demonstrates that C-A-S-H gels could grow on the surface of exogenous C-S-H seeds. It was also found that C-S-H seeds shorten the setting time of the slurry with a large volume of fly ash replacement, and the 14 d compressive strength of concrete added with 1% of the C-S-H seeds is similar to that of ordinary Portland cement [20]. Adding 2–4% of C-S-H seeds greatly increases the early strength of precast concrete admixtures, which could make the precast concrete complete within 24 h and increase the turnover efficiency of moulds [13,24,25].

The nucleation effect of C-S-H seeds in alkali-activated materials found to be controversial, depending on the chemistry of the activators [26,27]. The acceleration effect, due to the enhanced formation of C-A-S-H, is found in samples activated with hydroxides, while the decreased formation of the K-S-H gel, which induced deceleration in the silicates of the activated binder, was also noticed. It is also reported that incorporating hydromagnesite seeds accelerates the hydration rates and the degrees of the silicate-activated slag binder, as well as improves the mechanical performance by providing additional nucleation sites that enable the formation of more hydrates within the pores [28]. To the best of our knowledge, the effect of seeds on the carbonate-activated binder is rarely reported, which can be a good solution for the slow hydration of the binder.

Concerning the operability, ecologic footprint, and cost efficiency, carbonate-activated binders have gradually become a hotspot in the field of alkali-activated materials [1,4,7]. This study aims to understand the underlying mechanism of the C-S-H seeds in the carbonate-activated binders at the early age reaction and provide a solution that can promote the early age hydration and strength development of carbonate activation without weakening its operability and performance. The performance and microstructure of carbonate activation with C-S-H seeds are investigated in terms of reaction kinetics, phase composition, thermogravimetry (TG), and microstructure development at different curing ages, etc. Based on the obtained results, the reaction mechanism, the main influential factors of alkali-carbonate activation, and the seeding effect of C-S-H seeds are discussed. Finally, a fundamental reaction model to understand the early age hydration of carbonate-activated binders and the nucleation mechanisms is proposed.

## 2. Materials and Methods

### 2.1. Materials

Ground-granulated blast-furnace slag (GGBS) with a Blaine fineness of 437 m^2^/kg was used in this study (supplied by Shaoguan Steel Group Company Ltd. (Shaoguan, China)). The physical properties of slag were determined by a Mastersizer 2000 analyzer, Siemens/Bruker D5000 X-ray Powder Diffraction, and a FEI Quanta 450FEG Scanning Electron Microscope, and the results are shown in Figure 1, respectively. The oxide composition of slag was characterized using a PANalytical Epsilon3 analyzer, and the percentage of main components of CaO, SiO_2_, Al_2_O_3_, and MgO are 40.68%, 30.24%, 12.57%, and 9.05%, respectively.

The applied activator is the analytical grade sodium carbonate powder. The activator was first mixed with water and then cooled down to room temperature (20 ± 1 °C).

Calcium silicate hydrate (C-S-H) seeds with a Ca/Si ratio of 1.2 were prepared following the suggestion by [11,22], and their physicochemical compositions were characterized as shown in Figure 2. Next, 1 mol/L of sodium metasilicate nonahydrate solution and 1 mol/L of calcium nitrate solution were prepared with a 500 mL capacitor, respectively. Calcium sulfate solution, nitric acid, and sodium silicate solution were added to the container in proper order, with a controlled Ca/Si ratio of 1.2 and a water to solid ratio of 20. Next, 1–2 mL of NaOH solution was added to adjust the pH value of the solution to ≥12. The prepared solution was then cured in a continuous stirring pot at 80 °C for 5 d. Afterwards, the obtained suspension was repeatedly washed with deionized water 3 times. The solids, i.e., C-S-H seeds, were filtered out and then placed in a vacuum drying box for 2 d before experiments.

### 2.2. Experiments

The starting materials (Table 1) were firstly mixed and cast into moulds with the dimensions of 40 × 40 × 160 mm^3^. The samples were cured in conditions of R.H. ≥ 95% and a 20 ± 2 °C chamber before demoulding (depending on the hardening conditions). Afterwards, samples were cured at the same conditions until the ages of testing. The setting times and compressive strength of samples were carried out according to the standards of GB/T 1346-2011 (test methods for water requirements of normal consistency, setting time and soundness of the Portland cement) and GB/T17671-2021 (test method of cement mortar strength (ISO method)), respectively.

The reaction kinetics of the samples were characterized with an isothermal calorimeter (TAM AIR Calorimeter), set at 20 °C. The sample preparation took around 4~6 min (which was not recorded) and then the samples were loaded into the calorimeter. The presented results were normalized by the total mass of the sample. The samples at different stages were also characterized using a Siemens/Bruker D5000 X-ray Powder Diffraction (XRD) system, using a Cu tube (40 kV, 30 mA) with a scanning range from 3° to 55° 2θ and applying a step 0.02° and 30 s/step measuring time. The microscopic analysis was performed using a FEI Quanta 450FEG Scanning Electron Microscope (SEM). A Netzsch STA449F3 Jupiter^®^ thermal analyzer was used to measure the mass change. The samples were firstly held isothermally at 20 °C for 3 h to stabilize the internal balance of the TG instrument and then heated up to 1000 °C at a rate of 10 °C/min. Nitrogen was used as the carrier gas with a rate of 40 mL/min during the TGA measurements.

## 3. Results

### 3.1. Performance of Samples with C-S-H Seeds

Figure 3A shows the setting times of the sodium carbonate-activated slag binder with different dosages of C-S-H seeds. The reference sample shows long initial and final setting times of 225 min and 430 min, respectively. It is worth noting that the setting times of reference are shorter in comparison with previous research studies, whose hardening process of the carbonate-activated binders often takes around 3–7 d [5,11]. This is probably due to a false set, as indicated by the 1d-compressive strength of the reference sample. The setting times of samples are significantly shortened after dosing C-S-H seeds. The initial and final setting time of the slurry with 2% of C-S-H seeds reduce to 98 min and 244 min, respectively. Further increasing the dosage of C-S-H seeds results in a short setting time, and the initial and final setting time respectively decreased to 48 min and 118 min, proving that C-S-H seeds are an efficient agent in controlling early age hydration and the hardening process of SCAS binders.

The compressive strength of samples with different dosages of C-S-H seeds as a function of time is characterized as shown in Figure 3B. After 1 d of curing, the compressive strength of the sample with 4% of C-S-H seeds increases from no strength (not hardened) to 25.4 MPa. The 3 d-compressive strength of the reference reached 25 MPa, which is lower than the compressive strength of the samples added with C-S-H seeds. The 7 d-compressive strength of the reference sample develops rapidly, and the increase rate of the compressive strength of samples with C-S-H seeds turns slow. The 28 d-compressive strength of the sample with C-S-H seeds is similar to that of the reference, indicating that the incorporation of seeds mainly works on the early age of reaction.

### 3.2. Hydration of SCAS Binder with C-S-H Seeds

Figure 4A shows the isothermal calorimetric curve of samples with different C-S-H seeds. Similar to cement, the hydration of carbonate activation can be classified into five stages, i.e., preinduction, dormant, acceleration, deceleration, and steady stages. A long-dormant period of approximately 30 h for the reference sample was observed, which is in line with previous findings that the hardening process of SCAS may be long (depending on the chemistry and size of slag particles) [11,29]. The preinduction period of the hydration mainly occurs within 6 h of reaction, and the duration of the dormant period of the reaction is significantly shortened by adding C-S-H seeds. In comparison with the reference, the time to reach the exothermic peak of the specimen with 2%, 4%, and 6% of C-S-H seeds reduces to 13.5 h, 10.9 h, and 9.2 h, indicating that the precipitation of the main hydration products and the hardening process of the SCAS occurs earlier.

Seeding is a straightforward method used to control the early age reaction of cementitious materials, especially for samples with a high replacing level of cement [30]. The addition of C-S-H seeds acts as C-A-S-H gel nucleation sites, which increases the nucleation rate during the dormant period. The dormant period of the hydration is almost negligible after dosing C-S-H seeds, indicating that the incorporated seed controls the dominant period. The acceleration and deceleration periods of the reaction correspond to the second exothermic peak of the exothermic rate curve. The peak intensity of the exothermic peak is significantly stronger than that of the reference, indicating that the sample with the C-S-H seed crystals generates more hydration products at this stage. After 24 h of hydration, the reaction entered a stable stage with a low hydration rate, where the reaction rate was controlled by ion diffusion.

Figure 4B shows the cumulative heat release of samples within 72 h. After the consumption of carbonate anions reaches a certain level, hydration products, e.g., the C-A-S-H gel, began to precipitate and release heat. After adding C-S-H seeds, the total heat release of the slag slurry increased sharply at the first 24 h due to the main hydration, while that of the reference increases slowly. The cumulative heat release of the reference begins to increase significantly after 30 h of hydration, and the total heat release of the reaction to 72 h is slightly lower than that of the samples with C-S-H seeds. Based on the classical nucleation theory, the early hydration reaction rate of carbonate-induced cementitious materials is mainly controlled by the C-S-H nucleation growth, and the induction period is inversely proportional to the nucleation rate. The acceleration effect of C-S-H seeds on the early age hydration of the SCAS can be partially attributed to the seeding effect that provides more nucleation sites for the precipitation of the C-A-S-H gel, and the induction period is significantly shortened after dosing C-S-H seeds.

Figure 5 shows the SEM images of carbonate-activated slag with different dosages of C-S-H seeds after 1 d and 28 d of hydration. A loose structure with anhydrous slag particles is noticed for the reference sample after 1 d of curing, which is in agreement with the results that it does not gain strength after 1 d of reaction. The formation of the C-A-S-H gel is observed for the samples with C-S-H seeds cured for 1 d, and a relatively dense microstructure is found. Meanwhile, the flaky hydrotalcite and cubic calcite are also observed to be mixed with C-A-S-H gel, indicating the promoted early age hydration of the SCAS. With the increasing dosage of C-S-H seeds, the interior of the sample is filled with dense hydration products. After 28 d of hydration, the pores at the interface of the C-A-S-H gel and slag particle are filled with hydration products, and a denser structure is noticed for the reference sample and samples with C-S-H seeds.

### 3.3. Role of C-S-H Seeds on the Hydration Products

The XRD pattern of the 1 day cured sample of slag activated by sodium carbonate under different C-S-H seed content is characterized (Figure 6A). The broad diffraction peaks in the range of 25–35° 2θ come from anhydrate slag, indicating the presence of amorphous phases. The hydration products of the sample with C-S-H seeds after curing for 1 d are the C-A-S-H gel, hydrotalcite, hemicarbonate, vaterite, and calcite. After adding C-S-H seed crystals, the characteristic peak of the hydrotalcite phase at 23.4° 2θ and the broad peak at 29.3° 2θ of the low crystallinity C-A-S-H gel are observed to be significantly enhanced. A new polymorph of calcium carbonate, vaterite, is also observed for samples with a 4% addition of the C-S-H seeds. Vaterite is an unstable phase of calcium carbonate and will gradually transfer to other thermal stable phases of calcium carbonate, e.g., aragonite or calcite.

With an increase in the addition of C-S-H seeds, the peak intensity of hydrotalcite and hemicarbonate in the samples with C-S-H seeds increases gradually after 28 d of curing, exhibiting the continuous hydration of binders. However, the incorporation of C-S-H seeds did not affect the types of crystalline products in the long-term hydration, which remains as C-A-S-H gel and carbonate-containing phases. The broad diffraction peaks of the experimental samples at 29.3° 2θ also remains the same after 28 d of hydration, which could be due to the balanced mass change between the decreased amorphous phase in raw slag particles and the increased hydration products of the amorphous phase, e.g., C-A-S-H gel.

To further understand the role of the C-S-H seeds on the hydration products of SCAS, thermogravimetric analysis is performed to quantify the mass proportion (Figure 7). Generally, the decomposition peaks between 0 °C and 160 °C are attributed to the release of free water and the dehydration of C-A-S-H gel. The incorporation of C-S-H seeds promotes the nucleation and growth of calcium silicate hydrates, and the formed C-A-S-H gel loses bound water in the temperature range of 0 °C to 160 °C, as confirmed by the corresponding decomposition peak in the DTG (Figure 7B). The decomposition peak in the temperature range of 160 °C to 530 °C and 620 °C to 730 °C is attributed to the thermal decomposition of hydrotalcite and calcite, respectively.

Table 2 shows the mass loss of hydration products in different temperature ranges. To assess the extent of the C-A-S-H gel formation after dosing the C-S-H seeds, the bound water introduced by C-S-H seeds must be subtracted. The introduced C-S-H seeds considerably promote the early age precipitation of hydration products, including both C-A-S-H gel and carbonate-containing phases. The highest percentage of the newly formed C-A-S-H gel is found in the sample with 4% of C-S-H seeds, whose 1d-compressive strength is also the highest among all samples, indicating the contribution of C-A-S-H gel on the early age mechanical performance. With the increasing dosage of C-S-H seeds from 2% to 6%, the mass loss caused by the decomposition of hydrotalcite respectively increased by 146%, 156%, and 167%, compared with the reference sample.

## 4. Discussion

Accelerated hydration is observed for samples with C-S-H seeds. To further understand the role of C-S-H seeds on the early age hydration of carbonate activation, the hydration is divided into different stages based on reaction kinetics (Figure 4A), and the hydration products of samples with 4% of C-S-H seeds are characterized in terms of phase analysis, TG-DTG, and FT-IR every two hours (Figure 8). As can be seen, the incorporation of C-S-H seeds promotes the precipitation of carbonate-containing phases, especially at the first 2 h of reaction. Infrared spectroscopy also shows that the formation of C-S-H is promoted, as indicated by the vibration of bands at around 947 cm^−1^. It is noticed that the formation of the carbonate-containing phases (e.g., calcite, gaylussite, and hydrotalcite) occurs at the beginning of hydration, whose precipitation also helps the pH development of pore solution and dissolution of slag particles.

C-S-H seed is a pH-neutral compound and will not directly increase the pH values of the pore solution. Due to the high surface energy and large specific surface area of the C-S-H seeds, there is a large number of nucleation sites in the system. The incorporated C-S-H seeds allows the hydration products to be precipitated at a low saturation level, especially the carbonate-containing phases and strength-giving phase C-A-S-H gel. The acceleration effect of C-S-H seeds on the hydration of the SCAS should be related to its seeding effect that helps the precipitation of hydration products at the early age.

According to the nucleation theory, the nucleation process needs to overcome the nucleation energy barrier △G_r*_. When the seeds are introduced, the heterogeneous nucleation energy barrier is related to the wetting angle. Since the seed crystal is close to the lattice of the target product, the wetting angle is close to zero, significantly lowering the nucleation energy barrier of the target crystal form.
(1)$$△G_{r^{*}}=4/3\pi {r*}^{2}\sigma$$
(2)$$△G=△G_{r*}\cdot f\left(\theta \right)$$
(3)$$f\left(\theta \right)=1/4\left(2+\cos\theta \right){\left(1-\sin\theta \right)}^{2}\leq 1$$
where r* is the critical nucleus radius, σ is the surface free energy per unit area, and θ is the surface wetting angle between the nucleation phase and the seed crystal.

C-S-H seeds have similar crystalline structures to the C-A-S-H gel and can act as the preferred nucleation sites for C-A-S-H gels due to their high molecular recognition, greatly reducing the energy barrier to be crossed for nucleation. With an increase in the nucleation rate, the induction period of the hydration reaction shortened considerably. The addition of the C-S-H seeds simultaneously promotes the formation of hydrotalcite precipitation. In the slag glass body, SiO_4_^2−^ is a network former, and Ca^2+^ is a network modifier. A large amount of C-A-S-H gel precipitation continuously consumes elements such as Si and Ca in the solution, which promotes the dissolution of the slag. The dissolution of slags also accelerates the ionic release of Mg^2+^ and Al^3+^ and promotes the ionic strength development that raises the saturation limits of the pore solution. Consequently, carbonate-containing phases, such as hydrotalcite and calcite, begin precipitation at the early reaction. The hydrotalcite and C-A-S-H gel in the sample are mutually cemented to form the skeleton structure of the carbonate-activated gelling material, and the early performance of the sample is enhanced.

The C-S-H seed crystal acts as the preferred nucleation site, reducing the nucleation energy barrier of the C-A-S-H gel and increasing its nucleation rate. Compared with the other methodologies, the addition of C-S-H seeds can not only improve the early mechanical properties of the samples, but also ensure that the compressive strength does not decrease significantly in the later stage. The fluidity results show that the addition of the C-S-H seed slurry has better flow properties (not shown), which is beneficial to reduce the number of capillary pores in the matrix. Relevant studies also suggest that C-S-H seeds have better dispersibility and a more uniform microstructure of the hardened bodies compared with other nanoparticles.

Based on the obtained results, the role of the C-S-H seeds on the early age reaction of carbonate-activated binders is proposed (Figure 9). The dissolution of slag particles begins at the time when it mixes with the activator (Na_2_CO_3_), and soon the precipitation of calcium carbonate occurs. The consumption of CO_3_^2−^ anions raises the pH value of the pore solution that allows the ionic release of Si, Al, and Mg from the glassy phase of slags, increasing the ionic strength of the pore solution. According to the theory of classical nucleation, the acceleration mechanism of the C-S-H seed on the early age reaction of carbonate-activated binders is that the precipitate preferentially grows on the added C-S-H seeds. The nucleation of new phases overcomes less free energy due to the introduction of the nucleus, and the free energy ΔG is positively associated with the saturation level of the pore solution. In comparison with other seeds, the C-S-H seed shares a similar chemical structure to the C-A-S-H gel, the main strength-giving phase of the carbonate-activated binders, which facilities the formation of the C-A-S-H gel and carbonate-containing phases at the early reaction stages and promotes the hardening process.

## 5. Conclusions

This study investigated the early age reaction of carbonate-activated blast furnace slag and the effect of C-S-H seeds on the hydration was discussed. The reaction kinetics, hydration products, pH value of pore solutions, and ionic conditions were characterized. Based on the acquired results, the seeding effect of the C-S-H seeds was discussed. The following conclusions can be drawn:

(1) Synthesized C-S-H seeds are an effective accelerator for carbonate-activated slag that promotes the early age hydration and shortens the initial setting and final setting time of the sample. A higher dosage of C-S-H seeds leads to shorter setting times of the carbonate activation;

(2) The incorporation of C-S-H seeds allows the hydration products, e.g., the C-A-S-H gel and carbonate-containing phases, to precipitate at a lower saturation level, which controls the dormant period and increases the intensity of early age reaction. The dormant period of carbonate-activated slag becomes negligible;

(3) C-S-H promotes the early-age strength development of carbonate-activated slag, and its 1d-compressives strength increases from not hardened (reference) to 25.4 Mpa, which can meet the requirements of most engineering applications.

## Figures and Tables

**Figure 1 materials-16-01394-f001:**
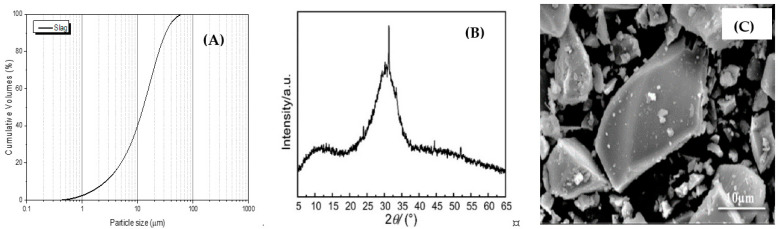
Physicochemical properties of slags: (**A**) particle size distribution, (**B**) XRD pattern, and (**C**) SEM picture.

**Figure 2 materials-16-01394-f002:**
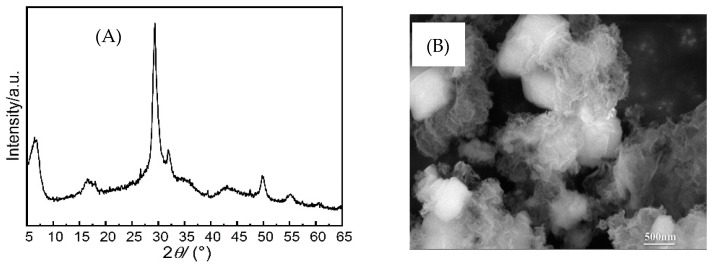
Properties of synthesized C-S-H seeds: (**A**) XRD pattern and (**B**) SEM picture.

**Figure 3 materials-16-01394-f003:**
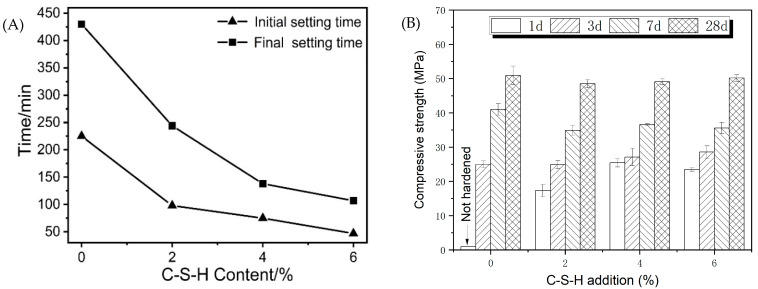
Properties of sodium carbonate-activated slag as a function of C-S-H seeds’ dosage: (**A**) setting times and (**B**) compressive strength.

**Figure 4 materials-16-01394-f004:**
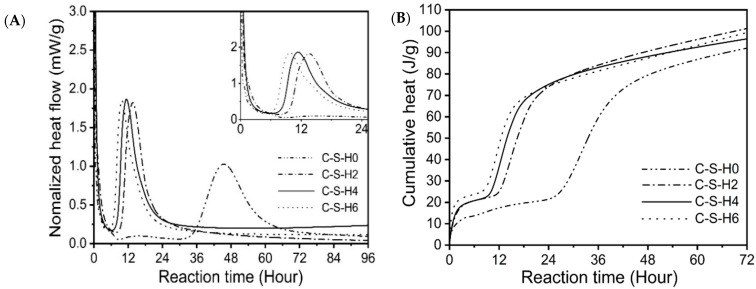
Heat release of samples with different dosages of C-S-H seeds: (**A**) normalized heat flow and (**B**) cumulative heat.

**Figure 5 materials-16-01394-f005:**
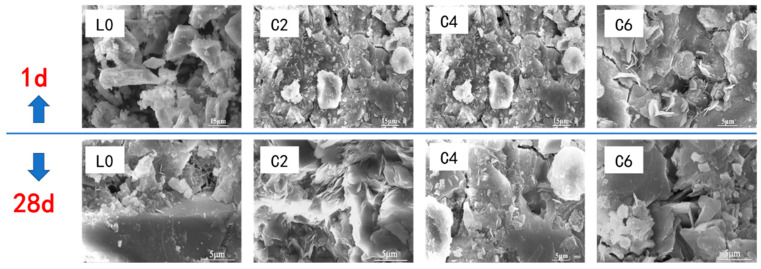
Microstructure morphologies of samples with different dosages of C-S-H seeds after 1 d and 28 d of reaction.

**Figure 6 materials-16-01394-f006:**
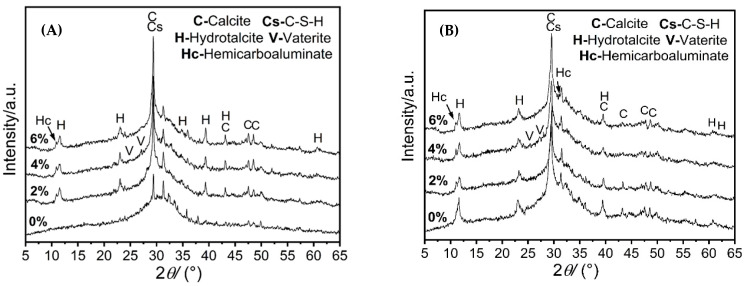
Phase characterization of samples with C-S-H seeds at different curing ages: (**A**) 1 d and (**B**) 28 d.

**Figure 7 materials-16-01394-f007:**
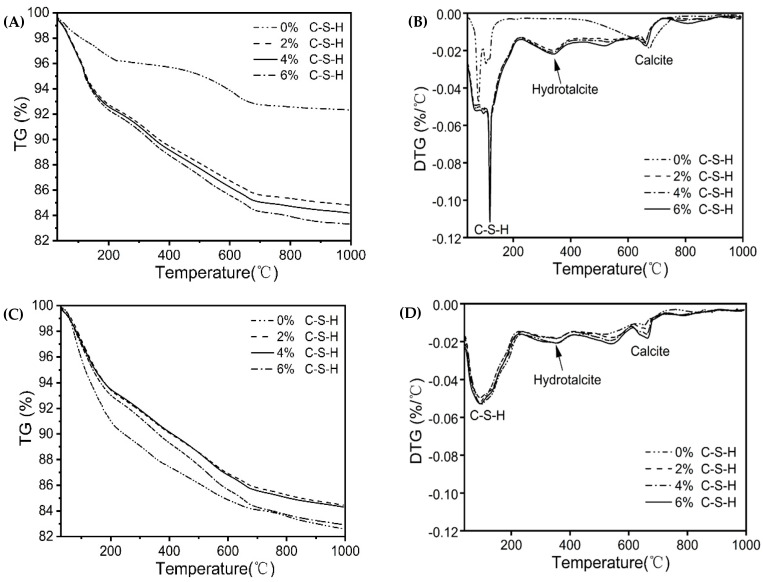
Decomposition of samples with C-S-H seeds as a function of temperature: (**A**) mass loss after 1 d of hydration, (**B**) derivative thermogravimetry after 1 d of hydration, (**C**) mass loss after 28 d of hydration, and (**D**) derivative thermogravimetry after 28 d of hydration.

**Figure 8 materials-16-01394-f008:**
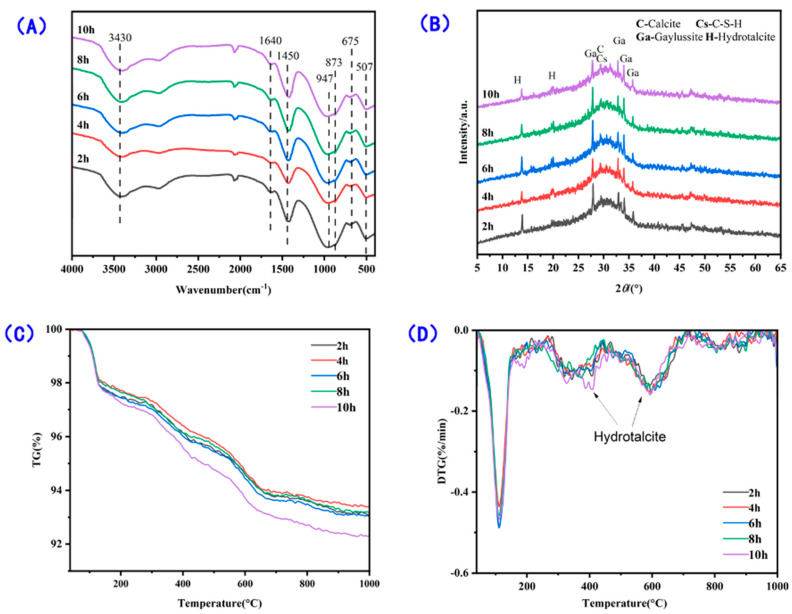
Early age hydration of samples with 4% of C-S-H seeds as a function of time: (**A**) infrared spectroscopy, (**B**) phase characterization, (**C**) mass loss at different temperatures, and (**D**) derived thermal gravity of samples.

**Figure 9 materials-16-01394-f009:**
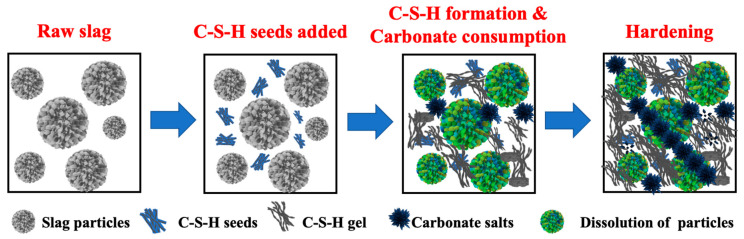
Proposed role of C-S-H seeds on the early age reaction of carbonate activation.

**Table 1 materials-16-01394-t001:** Deigned mixtures with different dosages of C-S-H seeds.

Mixture	GGBS	C-S-H Seeds Dosage	Na_2_CO_3_ Dosage	Water-To-Solid Ratio
L0	100	0	4%	0.4
C2		2%		
C4		4%		
C6		6%		

**Table 2 materials-16-01394-t002:** Mass loss of hydration products as a function of temperature after 1 d of curing (wt.%).

Temperature/°C	0–160	0–160	160–530	620–730
Decomposition of Products	C-A-S-H Gels and Free Water	Added C-S-H Seeds	Hydrotalcite	Carbonates
L0	2.78	0	2.43	0.69
C2	6.05	0.27	5.97	0.95
C4	6.42	0.54	6.23	1.00
C6	6.23	0.80	6.48	1.11

## Data Availability

Not applicable.

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
