# Peer review of "C-S-H Seeds Accelerate Early Age Hydration of Carbonate-Activated Slag and the Underlying Mechanism"

_materials, 2023, doi:10.3390/ma16041394_

Round 1

Reviewer 1 Report

Paper ID: materials-2166978

Type: Article 
Title: 
C-S-H seeds accelerate early age hydration of carbonate-activated slag and the underlying mechanism

Authors: Bo Yuan , Hengkun Wang , Dianshi Jin , Wei Chen

This study investigates an acceleration method for the eearly-agehydration of carbonate-activated slag by incorporating C-S-H seeds and unveils the underlying mechanism. Although the testing methods and compared results attained in the present study show the importance of the paper, The authors should address the following comments: 

  1. Novelty in comparison to recent literature? Need to be emphasized.
  2. The results in the paper might be discussed more by the relevant literature.
  3. There should be a space between the number and the unit. Please correct these errors in the paper.
  4. Line#32:  CLDHs?
  5. What is the difference between C-S-H and C-A-S-H? Please explain.
  6. Please add Blaine fineness of GGBFS.
  7. Line#133. 40 * 40 mm*160 mm3? Please do not use “*” in a scientific paper. Please change it to “×” .
  8. For setting times, which standard did the authors use? Please add it.
  9. For compressive strength, which standard did the authors use? Please add it.
  10. Line# 236. “roel” should be “role”.
  11. Throughout the text, some typos must be eliminated.

Author Response

The authors thank the reviewers for his/her thorough reading and for his/her positive feedback and helpful comments. All of the suggestions from the reviewers are carefully considered and addressed in this revised manuscript.

Reviewer 2 Report

The submitted manuscript is about mechanism of C-S-H seeds on the carbonate activated binders at the early age reaction. The presented manuscript must be improved by the following comments. Therefore, i kindly ask authors to prepare a response letter point-by-point rebuttal and must be subjected to the manuscript as well, considering the following comments with sufficient explanations.

1)    The C-S-H seeds must be explained in the introduction to make it clear for readers. What it is exactly and about its compositions, the way of producing and so on!

2)     In the abstract and in the line 13, “calcium silicate hydrates” must be added before (C-S-H) seeds.

3)     Lines 81-83 don’t have any citation.

4)  Introduction part is not impressive, Therefore, after line 41, the dissolution of portlandite and belite (C2S) clinker from the latest literatures is also suggested to be reported to make a comprehensive introduction easier for readers to understand. The following literatures are suggested to use for reporting this part  https://doi.org/10.3390/ma15196716https://doi.org/10.3390/ma15041442.

5)    The future application of this study must be reported.

6)    The quality of following Figures 1, 2, 3, 4, 6, 7 is not good. Possibly increase the resolution and vertical and horizontal’ axes titles of figures.

7)    In fact, heterogenous nucleation is more favorable (lower free energy barrier) than homogenous nucleation; therefore, during dissolution-precipitation hydration reactions at the very early ages for the low degree of supersaturation, heterogenous nucleation takes place. On the contrary, you mentioned that C-S-H seeds can help to the heterogenous nucleation. (lines 302, and 303). How you justify it?

8)     Please make more comprehensive conclusion as in the revised version the following points must be included; materials and methods, the significant of this study, the scope of the effort, the procedures used to execute the work, and the major findings.

Author Response

(The authors gave the same response as above.)

Round 2

Reviewer 2 Report

Perfect and it is accepted in present form.